Rhea: a transparent and modular R pipeline for microbial profiling based on 16S rRNA gene amplicons

Lagkouvardos Ilias ilias.lagkouvardos@gmail.com
Fischer Sandra
Kumar Neeraj
Clavel Thomas clavel.tom@gmail.com
ZIEL—Core Facility Microbiome/NGS, Technical University of Munich , Freising , Germany
Landa Blanca B.
Electronic publication date: 2017 Jan 11
Publication date: 2017
Volume: 5
Electronic Location ID: e2836
Received 2016 Jul 29; Accepted 2016 Nov 28
Copyright: ©2017 Lagkouvardos et al.
Copyright year: 2017
Copyright holder: Lagkouvardos et al.
License: This is an open access article distributed under the terms of the Creative Commons Attribution License, which permits unrestricted use, distribution, reproduction and adaptation in any medium and for any purpose provided that it is properly attributed. For attribution, the original author(s), title, publication source (PeerJ) and either DOI or URL of the article must be cited.
License URL: https://creativecommons.org/licenses/by/4.0/

Keywords: 16S rRNA gene, Microbiome analysis, R script, Microbial profile, Host-microbe interactions, Statistical analysis, Alpha-diversity, Beta-diversity, Correlations

Funding: German Research Foundation (DFG) Technical University of Munich (TUM) This work was supported by the German Research Foundation (DFG) and the Technical University of Munich (TUM) in the framework of the Open Access Publishing Program. The funders had no role in study design, data collection and analysis, decision to publish, or preparation of the manuscript.

==============================
The importance of 16S rRNA gene amplicon profiles for understanding the influence of microbes in a variety of environments coupled with the steep reduction in sequencing costs led to a surge of microbial sequencing projects. The expanding crowd of scientists and clinicians wanting to make use of sequencing datasets can choose among a range of multipurpose software platforms, the use of which can be intimidating for non-expert users. Among available pipeline options for high-throughput 16S rRNA gene analysis, the R programming language and software environment for statistical computing stands out for its power and increased flexibility, and the possibility to adhere to most recent best practices and to adjust to individual project needs. Here we present the Rhea pipeline, a set of R scripts that encode a series of well-documented choices for the downstream analysis of Operational Taxonomic Units (OTUs) tables, including normalization steps, alpha- and beta-diversity analysis, taxonomic composition, statistical comparisons, and calculation of correlations. Rhea is primarily a straightforward starting point for beginners, but can also be a framework for advanced users who can modify and expand the tool. As the community standards evolve, Rhea will adapt to always represent the current state-of-the-art in microbial profiles analysis in the clear and comprehensive way allowed by the R language. Rhea scripts and documentation are freely available at https://lagkouvardos.github.io/Rhea.

Introduction

The analysis of complex microbial communities by high-throughput sequencing of 16S rRNA gene amplicons has become very popular. However, this rapid popularization is in stark contrast to the slow transfer of knowledge about data analysis. The hope generated by pioneering next-generation sequencing studies community was high, but recent findings have highlighted the needs for standardization in the field (Clavel, Lagkouvardos & Hiergeist, 2016; Hiergeist, Reischl & Gessner, 2016; Sinha et al., 2015; Walker et al., 2015). With respect to data processing, important parameters such as the choice of clustering methods and thresholds of relative abundances must be considered for proper analysis of high-throughput 16S rRNA gene amplicon datasets (Edgar, 2013; Martínez, Muller & Walter, 2013). The common output of raw sequence processing is a contingency table with read counts assigned to sequence clusters (usually at 97% similarity) termed Operational Taxonomic Units (OTUs). These tables are commonly referred to as OTU-tables and also contain information on the taxonomic classification of sequences. There are several options for the processing of raw sequence data down to OTUs-tables: the most popular software (or package thereof) are QIIME (Caporaso et al., 2010), mothur (Schloss et al., 2009) and UPARSE method (Edgar, 2013) offered by the USEARCH package (Edgar, 2010). The UPARSE method has also been integrated into the pipelines LotuS (Hildebrand et al., 2014) and IMNGS (Lagkouvardos et al., 2016) that can also be used for initial processing of raw sequencing data. In particular, IMNGS is a web-based platform (http://www.imngs.org) with no installation requirements.

Downstream analysis of OTU-tables is carried out for the calculation and visualization of diversity and composition of complex microbial communities of interest. Further statistical analysis can allow the identification of OTUs or taxonomic groups that are differentially abundant between sample groups, or can highlight correlations between microbial taxa and metadata characterizing the environment of interest. OTU-tables and associated sequences can currently be analysed by specialized software suites like QIIME (Caporaso et al., 2010) and mothur (Schloss et al., 2009), or by individual pipelines using a general statistical platform like R (R Core Team, 2013). As it is challenging to establish analysis environments that satisfy the needs of a wide diversity of users with respect to ecosystem of interest, experimental design, and expectations, these specialized tools with their respective strengths will continue contributing substantially to standardized analysis of amplicon datasets. Nevertheless, there is also room for further development of modular, transparent, and user-friendly approaches.

In contrast to specialized software suites, usage of open statistical environments like R is flexible. The broadness of available tools and the simple syntax can quickly accommodate very complicated needs and allow adjustment to novel requirements, reducing the need for dedicated programs. That is why the R environment has established as a very versatile, powerful, and comprehensive platform for the development of analysis pipelines. The only drawback is the scripting required that can discourage new R users. Pre-packaged functions dedicated to the analysis of microbial data already exist in R, including the phyloseq package (McMurdie & Holmes, 2013), greatly reducing the required personal implementation. Furthermore, the development of Shiny-phyloseq (McMurdie & Holmes, 2015), which adds an abstraction layer over the R console with a browser based graphical control of the analysis, is minimizing the need for knowledge about the R language.

Here we took advantage of the easiness and flexibility of R to create a full assembly of analytical steps for diversity and composition analysis of OTU-tables. These scripts were optimized for the analysis of host-associated microbial profiles, most specifically those from the gut environment. We have already utilized the pipeline in different studies (Lagkouvardos et al., 2015; Schaubeck et al., 2015; Würth et al., 2015), which speaks in favour of its validity and usefulness for the assessment of relevant biological information from 16S rRNA gene amplicon data. These scripts do not represent an exhaustive set of all possibilities that exist to analyse sequence datasets, but rather reflect our personal choices among available methods together with novel ideas integrated in the selected analytical steps. We believe that this pipeline can be of use to many scientists with limited or no programing experience or to those with more experience who wants to build on an existing framework, thanks to a simple and clear step-wise processing with very detailed documentation. We packaged this pipeline under the name of the ancient Greek mythology goddess “Rhea,” as a reference to its ever flowing modular architecture, and make it freely available at https://lagkouvardos.github.io/Rhea.

Methods, Results and Discussion

Overview

The Rhea pipeline consists of six main R scripts that perform the tasks of normalization of input tables, calculation of alpha-diversity, beta-diversity, taxonomic relative abundances, serial group comparisons, and correlations. These scripts rely on the R packages ade4, GUniFrac, phangorn, Hmisc, corrplot, plotrix, PerformanceAnalytics, reshape, ggplot2, gridExtra, grid, ggrepel, gtable, and Matrix that by themselves have several dependencies. The installation of the packages is performed automatically within the scripts when run for the first time. For the purpose of demonstrating and illustrating the different features of Rhea, the publicly available sequence data from the study by Müller et al. (2016), with ENA accession PRJEB13041, were analysed with the web platform IMNGS (http://www.imngs.org) and the output OTU-table and files (also available for download through the GitHub repository) were used for analysis. In this template study, the impact of housing conditions and diet on mouse faecal microbiota and gut barrier was investigated. For demonstration of the variability of OTU-specific relative abundances among technical replicates, a small study of 10 amplicon libraries constructed from the same human faecal sample were sequenced by Illumina MiSeq. The raw data of this study were deposited to ENA and are available under accession PRJEB14963. Those data were also processed with the web platform IMNGS and the OTU-table and all intermediate steps for the analysis of Coefficient of Variation can be find in Table S1.

The following sections describe in detail each of the main functions of Rhea, thereby emphasizing on important concepts underlying data processing using the scripts. Meticulous documentation of all scripts is provided online at the link given above in the abstract. To minimize manual handling of data, intermediate files generated during processing are automatically transferred to folders where they are needed for downstream analysis, on the condition that the original folder structure of Rhea is kept unchanged. Illustrations shown in the present manuscript correspond to raw outputs generated by Rhea, with very minor post-production manual changes (i.e., size and orientation may have been changed to facilitate publishing).

Normalization

This is the first script to be used, as normalization of OTU-tables is a prerequisite for any downstream analysis. Normalization is the process of transforming data to remove confounding effects of different sample sizes. As sequencing usually results in different number of sequences per samples, a normalization of read counts is required prior to downstream analysis. This is commonly performed by a procedure called rarefying, i.e., a random sub-sampling of the reads from each sample to a fixed total (usually the least count among the samples). Rarefaction, although very popular among ecologists and microbial ecologists, has been criticized for the following reasons: (i) omission of available valid data, (ii) the estimation of over-dispersion is more difficult due to data loss, (iii) loss of power (type II error), (iv) dependence on an arbitrary threshold, and (v) additional uncertainty due to the randomness in rarefaction (McMurdie & Holmes, 2014). The authors of the latter publication stated that even a simple normalization to proportions is less biased as it includes no random steps and minimal loss of information. Their suggested normalization consisted of a variance stabilization transformation (logarithmic). In Rhea, the issue with this kind of transformation is the incompatibility with some of the downstream analytical functions requiring counts or proportions. Plotting of relative abundances across groups expressed in percentages, with its known limitations, gives researchers a more intuitive understanding of biological phenomenon. Hence, in Rhea, counts are by standard normalized via simple division to their sample size and then multiplication by the size of the smaller sample. This approach has not the downside of introducing random variance or loss of data. Nevertheless, we provide to users the option to proceed with classical rarefaction for normalized counts if wanted. For proper comparison analysis, it is important that all samples have initially similar sequencing depth (Bálint et al., 2016). If for example a sample is represented by 2,000 reads and all others by approximately 20,000, it is recommended that the shallow sequenced sample is removed. This is not only due to the bias introduced by normalization when dealing with grossly unequal depths, but also because such differences are indicators of experimental failure. Errors in DNA purification, quantification, PCR amplification, barcoding, and most of all sequencing can all lead to low sequence outputs, and downstream analysis is not meant to compensate for these problems. In the case of our template dataset, all samples had sufficient and similar coverage and were kept. Finally, although the use of negative controls during sequencing is strongly recommended to help estimate artefacts and identify contaminations, these negative control samples should be removed during final analysis of target samples as their very low number of reads will dramatically affect results in downstream analyses. The script in Rhea does not automatically remove those samples: it is the responsibility of users to control the quality of input OTU-tables. If processing of raw sequences was done using IMNGS (http://www.imngs.org), we recommend repeating the analysis twice: once with the control samples to possibly identify important spurious OTUs, and once only with the target samples for downstream analyses. By running the Normalization script, four output files are created and copied automatically to the folders where they are needed for next steps.

Alpha-diversity

The diversity of OTUs within a given sample is the alpha-diversity of that sample. The simplest way of measuring it is to enumerate OTUs present in that sample, also called species richness. In Rhea, since we use normalized counts, we consider only those OTUs that occur above the threshold of 0.5 counts (i.e., the risk that OTUs below 0.5 counts appear due to disproportional sequencing depth is high). Species richness does not consider the structure of communities and does not adjust for differential abundance of individual OTUs. In an analogy to species richness measurement of alpha-diversity, a school with one female and 99 male students would be as diverse as a school having a 50/50 ratio over the genders. There are different indices that can capture the community structure rather than enumerating the parts. The two most popular are the Shannon (2001) and the reverse Simpson (1949) diversity indices, with the later adding more weight to highly abundant taxa. Those indices are not linear, meaning that a sample with Simpson index of 0.7 is not twice as diverse as a sample with Simpson index of 0.35, which can lead to misleading illustrations and thus interpretations as well as inappropriate statistics. In addition, diversity indices are not measures of the actual diversity, but rather proxies for understanding that diversity, in the same way as the radius of a sphere is not a measure of its volume but an index for it. These limitations have been analysed in depth by Lu Jost, who proposed the use of effective diversity (Jost, 2006; Jost, 2007). In short, effective diversity of a sample with an index value X is the number of equally abundant species that would give the same value for that index. In Rhea, we calculate both the Simpson and Shannon indices and their effective numbers. Besides species richness, we recommend using only the effective diversities for visualizations or for statistical comparisons across samples. The results from calculation of the alpha-diversity for the template samples are available in Table S2.

Beta-diversity

The diversity of OTUs across samples is called beta-diversity. This is done by applying a distance metric over their taxonomic or genomic profiles that result in an all-against-all distance matrix. There are multiple ways to calculate distances between samples based on similarity of their members: the most common are the Bray-Curtis and the weighted and unweighted UniFrac distances (Bray & Curtis, 1957; Lozupone et al., 2007). While Bray-Curtis only considers the shared taxonomic composition across samples, UniFrac takes into consideration the genetic distance of the community members (OTUs) in each sample to the members in the other samples. Weighted UniFrac adds information about the relative abundance of each OTU to every genetic distance. Because unweighted and weighted UniFrac are sensitive to rare and dominant OTUs, respectively, a balanced version was proposed, referred to as generalized UniFrac (Chen et al., 2012). In Rhea, we use the generalized UniFrac for calculation of the phylogenetic distance matrix.

The next step is the visualization of the generated distance matrices in a perceivable space usually of two or three dimensions. This is achieved by either Principal Components Analysis (PCA) or Principal Coordinates Analysis (PCoA), the latter being also known as Multi-Dimensional Scaling (MDS). In Rhea, we calculate both MDS plots based on the samples distances and the non-metric version of MDS (NMDS). Because the latter is commonly regarded as the most robust unconstrained ordination method in community ecology (Minchin, 1987), we recommend its usage. A PERMANOVA test (vegan::adonis) is performed in each case to determine if the separation of sample groups is significant, as a whole and in pairs (Anderson, 2001). In addition, a dendrogram is produced from all the samples hierarchical clustered using the Ward’s clustering method (Murtagh & Legendre, 2014). For the template dataset, the PERMANOVA test showed that faecal microbial profiles of mice fed control diets separated significantly (p-value < 0.001) from that of mice fed high-fat diets (Fig. 1A), for which additional effects of the housing facility (hygiene) were observed (p-value = 0.012 for the pairwise comparison). Details on sample clustering can be seen in the produced dendrogram (Fig. 1B).

Figure 1 Representative graphical output of the Beta-Diversity script.

(A) NMDS plot of generalized Unifrac distances showing distribution of the mouse fecal samples based on phylogenetic makeup of their microbiota. The plot shows a significant separation of sample groups according to diets, and facilities in the case of the high-fat diet (pairwise statistics are also generated by Rhea in a different output file). (B) Dendrogram output of the Beta-Diversity script showing the distance and clustering of individual samples based on the Ward’s clustering method. Abbreviations: CD, control diet; CON, conventional facility; HFD, high-fat diet; SPF, specific pathogen-free facility.

Taxonomic binning

For every taxonomic level, the relative abundances of all OTUs sharing the same taxonomy are summed. This agglomeration of relative abundances is calculated also for sequences of unknown taxonomic placement but otherwise sharing the same root. Stacked relative abundance bar plots are produced for all samples at any given taxonomic level. Although this graphical representation may help to quickly visualize overall composition differences (Fig. 2), rigorous statistical analysis of differential relative abundances at different taxonomic levels across samples must be performed to identify bacterial groups that may vary according to the specific condition under investigation.

Figure 2 Representation of the bacterial composition in each sample as obtained by the Taxonomic Binning script.

It can be seen immediately that the bacterial phyla (p) Bacteroidetes and Firmicutes dominate the communities in all samples. An increased sequence proportion of Firmicutes in the samples from mice fed the high-fat diet compared with mice fed the control diet can also be observed, although we strongly recommend using the Serial Group Comparisons script for proper analysis and illustration of differences.

Serial group comparisons

A common objective of microbial profile analysis is to compare compositions between groups of samples sharing a specific feature. This can be done by performing an Analysis of Variance (ANOVA). As a parametric test, classical ANOVA assumes normality of data distribution. Since this is rarely the case for OTU-based data, we use the non-parametric Kruskal–Wallis Rank Sum Test in Rhea (Hollander, Wolfe & Chicken, 2013). In cases where only two groups are compared, ANOVA is equivalent to a t-test, or in Rhea to a Mann–Whitney test for non-parametric data. When more than two groups are compared, pairwise tests are needed to determine the groups that are significantly different. Again, the non-parametric Mann–Whitney Test (Anderson, 2001) is used therefore. The obtained overall and pairwise significance values are corrected for multiple testing with the Benjamini–Hochberg method (Benjamini & Hochberg, 1995) and are reported for each variable in the graphical output of the analysis.

Rhea was designed to perform systematic testing of all available OTUs or taxonomic groups in a given experiment. This results in many tests to be performed in series and is thus associated with a high trade-off in adjustment for false discovery rates. To avoid comparisons of taxa that may not be representative in the given ecosystem of interest (e.g., very scattered occurrence), we strongly encourage thorough knowledge of input data to be able to identify thresholds that can be set in Rhea to avoid analysis of those low relevant taxa, as it was shown that pre-filtering data increases the power of analysis (Bourgon, Gentleman & Huber, 2010).

The first step in the present script is transformation of the OTU or taxonomic relative abundances table by replacement of near-to-zero values with zero. The rationale is based on both the high uncertainty of taxa with low relative abundances and the incentive to identify samples where an organism showing significant differences is clearly an important component of the communities. The default cut-off in Rhea is 0.5%, i.e., all relative abundances of any taxa in any sample below that threshold will be considered as absent. Note that this is based on our own experience in analysing gut-derived microbial communities, and that this cut-off can be easily changed or even deleted for cases where abundances of rare OTUs are deemed important. The rationale behind this transformation is the usually high variation characterizing taxa with low relative abundances, which means that those taxa can appear or disappear randomly across sequencing replicates of the same sample (Fig. 3). Therefore, by zeroing them consistently we reduce the noise and help distinguishing samples where the given organism is considered to be present with a confident degree of certainty.

Figure 3 Relative sequence abundance of OTUs (x-axis) is negatively associated with the coefficient of variation (y-axis) calculated from replicate samples.

The analysis included 10 libraries of 16S rRNA gene amplicons constructed from the same human faecal sample and sequenced on 4 different dates using 2–4 replicates per sequencing run. Each blue circle represents the mean value of the 10 replicates for a given OTU. It can be seen that most OTUs with a mean relative abundance <0.1% are characterized by coefficient of variations (CV) >30%, illustrating substantial variations between replicates. OTUs with relative abundance between 0.1% and 0.5% show intermediate CV, with a mean of 30.72%. The default threshold used in many steps in Rhea to filter datasets is 0.5% relative abundance. Although most OTUs above this cut-off (n = 25) were characterized by low mean CV of 15.56%, some (n = 7) did show CV between 40 and 80%, which illustrates reproducibility issues pertaining to high-throughput sequencing, at least in the context of the present study (additional data by others will be needed prior to generalization of these results). Original raw data are deposited in ENA (PRJEB14963). The OTU table and all intermediate steps followed for production of the figure can be found in Table S2.

The second important transformation is the exclusion of zeros as true values in OTU and taxonomic abundance tables (and their replacement by the Non-Available (NA) notation). We consider a value for a given OTU (or taxonomic group) for statistical purposes only when it can indeed be detected. The fact that a taxon is not detected by sequencing does not necessarily mean that it is completely absent in the corresponding sample; it can be present at a population density that is too low for the detection limit of the method. Hence, including zero values in the analysis distorts artificially the distribution of values. Moreover, the conditions that we commonly investigate for differential abundances are multifactorial. For example, high relative abundances of certain organisms can be associated with a disease, but disease alone is not always indicative of the existence of a specific organism. In those cases, usage of zeros can mask actual biological signals and hinder the detection of risk factors (Fig. 4). Therefore, to avoid these pitfalls, we recommend treating zeros as missing values during statistical tests in Rhea.

Figure 4 Effect of excluding zero values before statistical tests.

Artificial data representing three hypothetical OTUs (tables on the left) were manually generated to visualize the effect of removing zero abundance values on data interpretation. The fictive scenario behind these data was that a disease phenotype was associated with higher relative abundance of any of these OTUs. This practically mean that absence or near-to-zero relative abundances of individual OTUs in a sample is not indicative of the absence of phenotype as soon as any of the other OTUs are present in sufficient amounts. Leaving zeros and near-to-zero values was masking this biological information giving no significant p-values for the comparisons (upper right block). When the transformation was applied, the fact that those three OTUs whenever “present” have higher median relative abundance in samples originating from diseased vs. healthy individuals was revealed (lower right block).

Zeroing values and treating them as missing data, as explained in detail in the previous sections, can substantially affect the number of positive samples within a group (prevalence) and thereby markedly influence the interpretation of data. For instance, although statistical testing may return a significant result, it would be misleading to conclude that a specific OTU generally occurs at a higher relative abundance in group A vs. B if the prevalence in A is 4 of 10 samples where the number of positive samples in B is 12 out of 12 (all samples are positive but the relative abundance of the given OTU in these samples is low). Hence, it is very important to take into consideration the sample prevalence for a given taxon showing significant differences between groups to avoid drawing possibly misleading conclusions. In Rhea, a prevalence-based filtering can be applied on OTU/Taxonomic variables prior to statistical testing. Per default, if there is not at least one sample group where the variable of interest is present in more than 30% of samples, then the variable is considered too sparse and is not tested. For example, in a case of two groups of each 10 samples, if an OTU or taxonomic group is not detected in at least four samples in any group, then it is not tested. This threshold can be easily adjusted in the parameters of the script. In addition, prevalence patterns are statistically tested with Fischer test (Fisher, 1950). Obtained p-values are corrected for multiple testing using the Benjamini–Hochberg method (Benjamini & Hochberg, 1995) and reported in tables below the respective plots.

Rhea offers one additional level of filtering. Significant differences in variables characterized by low relative abundance values across all groups are difficult to interpret with respect to their importance in the whole community (e.g., increase in median relative abundance from 0.25% to 0.70%). To avoid the burden of interpreting spurious associations, we only perform tests for variables with a median that is above a defined cut-off in at least one group of samples. The default cut-off is 1%, meaning that only variables that have at least one sample group characterized by a median relative abundance above 1% total reads will be considered for testing.

Finally, besides the filtering opportunities mentioned above, manual selection can be applied by removing certain categories of variables from the input file. For example, taking into account the high redundancy inherent to binning OTU counts through entire taxonomic lineages, we commonly restrict statistical analysis to phyla and families. Moreover, we do encourage users to carefully select the type of groups they want to compare, in order to limit comparisons over many different groups containing low numbers of samples. This would for instance be the case if samples are grouped according to too many variables at a time (e.g., a combination of treatment, genotype, time of measurement). All of the existing filters in Rhea are easy to adjust and a log file listing the selections made is created for every run of the script for future reference and for the sake of reproducibility.

As graphical outputs of the Serial Group Comparisons Script, Rhea offers three possibilities (box, violin, and dot plots) that can easily be set by the user in the upper modifiable part of the script. An example of these outputs is provided in Fig. 5.

Figure 5 Serial group comparisons graphical output (provided as pdf files).

The script produces three different representations of variables showing significant differences between the two groups as. (A) Dot plot of a meta-variable (concentration of the bile acid deoxycholic acid in mouse feces). This representation showing individual values is usefull when groups are represented by a low number of samples. (B) Box plot representation of the alpha-diversity between control and high-fat diet mice shown as the Shannon effective count of species. (C) Violin plot of an OTU showing significantly different median relative abundances between the two groups.

Correlations

For the detection of variables that show the same or reverse direction of changes across individual samples, a correlation calculation script is available in Rhea. This script makes one important separation across the variables provided in the input table. They are considered either as meta-variables (continuous measures of physicochemical variables from the samples) or as taxonomic variables (relative abundances of OTUs or higher taxonomic levels). This guarantees a better control over data transformation and the number of performed tests. By default, correlations between all meta-variables and taxonomic variables are calculated. If relevant, all pairwise correlations within meta-variables and/or taxonomic variables can be calculated and reported too.

Sequencing data (read counts, OTUs, and taxonomies) are compositional and therefore data transformation is needed before true correlations across variables can be detected (Aitchison, 1986). In simple terms, this refers to the artificial dependency of parts of a community when sampled and expressed as proportions. Independent changes in numbers of one member of a community affect the relative abundance of all other members even if their absolute numbers have not change. This can lead to misleading conclusions on correlations between relative abundances among members of the population. In Rhea, the centred log-ratio transformation is used to remove the compositionality constrains in taxonomic variables. Following this transformation, the table is centred and scaled, and the Pearson correlation for all pairs is calculated (Pearson, 1909). The significance of each observed correlation (also after correction) is reported together with the number of observations that supports it. As for the Serial Group Comparisons tests, selections of only some taxonomic levels for calculating the correlations can help avoiding repetitive testing of redundant data. Variables that appear in less than a fixed percentage of samples can also be removed from the analysis (30% is the default minimum number of samples that a taxonomic variable should appear in order to be tested for correlations). The removal of zeros and near-zero values (as explained in detail above) is also recommended for the taxonomic variables during correlations calculation, i.e., if an OTU is detected >0.5% relative abundance in 9 out of 30 samples, only the corresponding values will be used for calculation of correlations to other variables. Rhea always reports the number of observations (pairs of values) that supports each calculated correlation. The Correlation Script generates two graphical outputs shown in Fig. 6.

Figure 6 Graphical output of the Correlations script in Rhea. (A) Correlation array across all tested variables. The color of circles corresponds to the direction of correlations and their size to the significance of the p-values before correction (the bigger the circle, the lower the p-value and thus the higher the significance). (B) Examples of individual correlation outputs as generated by Rhea showing one strong positive (OTU_22 Unknown Lachnospiraceae) and one negative (OTU_32 Unknown Rhodospirillales) correlation extracted from the pairwise tests (the taxonomic classification of the corresponding two OTUs by RDP classifier were added manually). The correlation coefficient and the original and corrected p-values are shown on top of each plot, the number of observations supporting the correlations below the plot. The straight bold line correspond to the fitted linear model while the gray area within the dashed lines represents the confidence intervals for that model. Since correlations are calculated from transformed data, no units are shown.

Conclusion

The analysis of complex microbial communities by high-throughput 16S rRNA gene amplicon sequencing has become very popular. However, there is still a substantial gap between high usage needs and the limited expertise available in many users’ laboratories. In the present manuscript, we describe and make available a new resource that will help addressing this issue. Rhea is a transparent and modular assemblage of R scripts that allows fast and easy processing of OTU-tables to produce an array of most commonly used readouts in the field of microbial ecology.

Rhea can be used for analysis of amplicon datasets from all types of environment. Nevertheless, its development has been shaped by our own interest in studying communities colonizing the human and mouse intestine, which influenced the creation of certain parameter options. For instance, the intestine is characterized by a relatively high turnover due to bowel movements and constant renewal of the host epithelium and associated secretion (e.g., mucus). Combined with constant inflow of exogenous microbes from the environment via the oral route, and taking into account inherent detection of spurious OTUs in high-throughput datasets (Glassing et al., 2016; Martínez, Muller & Walter, 2013), it is important to carefully interpret results related to the occurrence of low abundant taxa, which are likely not to represent important populations of endogenous microbes. That is why we implemented filtering options in Rhea based on specific thresholds of relative sequences abundances. In combination with cut-offs on prevalence, these options also reduce the burden associated with false discovery rates when testing very high numbers of variables. Importantly, all parameters can be easily adjusted in the scripts according to the needs of experimenters. It is also important to note that, in the case of explorative studies (i.e., when the primary aim is to generate hypotheses and not to provide irrefutable proof of e.g., a specific association between the occurrence of microbes and a specific conditions), and because p-values adjustment is still a matter of debate, at least in clinical science (Feise, 2002), Rhea offers interpretation of data both before and after correction of p-values.

The rationale behind Rhea is really to share knowledge usable for analysis of 16S rRNA gene amplicon datasets by providing a ready-to-use transparent pipeline (both in terms of scripts structure and detailed documentation that explains choices made for analysis and how to use scripts). This transparency combined with easiness of implementation and a modular aspect makes Rhea useful for both novice and more-advanced users, in spite of already existing state-of-the-art tools for analysis of microbial profiles.

Supplemental Information

Table S1 Coefficient of Variation of OTU abundances across technical replicates

Click here for additional data file.

Table S2 Output of alpha diversity script

Click here for additional data file.

Additional Information and Declarations

Competing Interests

Author Contributions

DNA Deposition

Data Availability

The authors declare there are no competing interests.

Ilias Lagkouvardos and Thomas Clavel conceived and designed the experiments, performed the experiments, analyzed the data, contributed reagents/materials/analysis tools, wrote the paper, prepared figures and/or tables, reviewed drafts of the paper.

Sandra Fischer performed the experiments, analyzed the data, contributed reagents/materials/analysis tools, prepared figures and/or tables, reviewed drafts of the paper.

Neeraj Kumar performed the experiments, analyzed the data, contributed reagents/materials/analysis tools, reviewed drafts of the paper.

The following information was supplied regarding the deposition of DNA sequences:

ENA with accession PRJEB14963.

The following information was supplied regarding data availability:

GitHub: https://github.com/Lagkouvardos/Rhea.

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
