# Peer review of "Rhea: a transparent and modular R pipeline for microbial profiling based on 16S rRNA gene amplicons"

_PeerJ, doi:10.7717/peerj.2836_

## Round 0.1 · original submission · Major Revisions

Dear authors

Both of the reviewers indicated an important lack of the novelty in your study. However, PeerJ. considers that decisions for rejection are not made based on any subjective determination of impact, degree of advance, novelty, or for being of interest to only a niche audience. So, I cannot consider this as a lack of merit for your work. However, apart from that both reviewers encountered a substantial amount of details in the materials and methods and result sections that need to be amended. Specially, some of the interpretations concerning the work by McMurdie & Holmes are considered not being adequate. Additionally, there are many comments and aspects that need additional work and validation. I encourage you to take into account all comments driven by the reviewers and try to answer them in order to improve your manuscript.

Reviewer 1 ·

Basic reporting

See below

Experimental design

See below

Validity of the findings

See below

Additional comments

Lagkouvardos and colleagues present a manuscript describing an R package for analysis of OTU tables generated from 16S rRNA gene sequence data. It is unclear to me what advantages this package gives a user over other existing packages in the literature including base R, phyloseq, or the tidy-verse (e.g. dplyr, ggplot2, etc). Ultimately, it is not clear what this manuscript adds to the existing literature and analytical landscape.

1. The authors make a number of unsubstantiated claims that would be interesting to see backed with data. For example, "Rhea is primarily a straightforward starting point for beginners, but can also be a framework for advanced users who can modify and expand the tool" and "Restrictions of these existing approaches commonly expressed by users include difficulty of installation, complicated implementation for non-educated users, and demanding skills required for modification". I am not sure that the former is true of most R packages since the user still needs to learn a new syntax. For the latter, these have mostly been criticisms against QIIME, but would be interesting to see quantified.

2. The authors make the claim that "Rhea will adapt to always represent the current state-of- the-art in microbial profiles analysis in the clear and comprehensive way allowed by the R language". It is nice to hear the authors state this, but the fact is that most open source software packages are not maintained for more than a year or so. Considering this is a rapidly growing area of research and given the current development tradjectory for OSS, this claim comes off as hollow.

3. At L139 the authors do not clearly articulate what they mean by rarefaction and seem to perpetuate the misconception from the McMurdie & Holmes article. Furthermore, they fail to acknowledge that the McMurdie & Holmes position is very much in the minority of ecology and microbial ecology researchers. Rarefaction is the most robust method for controlling for uneven sampling size. It involves randomly selecting a fixed number of individuals from each sample, calculating a parameter (e.g. Shannon diversity or Bray-Curtis), and repeating a large number of times. The average parameter is then reported. It does not appear that this best practice is adhered to in this manuscript.

4. I believe that mothur should not be capitalized since it is not an acronym and was left uncapitalized in the original manuscript.

5. The authors' R package should be submitted to CRAN and preferably BioConductor to help insure that it remains publicly accessible and conforms to a specific set of programming standards.

·

Basic reporting

No Comments

Experimental design

No Comments

Validity of the findings

See general comments.

Additional comments

- General comments:
The authors present a post-clustering pipeline for analyzing OTU abundance counts in combination with other metadata, in order to generate near-publication-ready tables and figures in a newbie-friendly way. Each script is accompanied by a very clear and well-organized documentation file, which covers the theoretical background, use cases and common. Additionally, the authors present some novel ideas, such as excluding zero OTU count values before statistical testing, that are conceptually attractive and appear to improve the resolution of their pipeline – albeit the testing of that particular method was performed with manually generated data, so it is hard to assess its actual impact on real datasets.

I only have one major concern regarding the normalization step. The authors cite McMurdie & Holmes (2014) in order to support their usage of normalization to proportions in Rhea. However, to the best of my understanding, McMurdie & Holmes (2014) do in fact claim in their paper that neither rarefaction nor normalization to proportions are acceptable for analyzing OTU count data (see below), and instead propose alternative methods which are actually also implemented in R and would be easy to incorporate into the Rhea pipeline. The authors should either follow the recommendations of McMurdie & Holmes (2014), or explicitly discuss why they believe that normalization to proportions is enough for their purposes.

More detailed comments can be found below.


- L110: “The installation of the packages is performed automatically within the scripts when run for the first time.”

The automatic installation of unmet dependencies should indeed be of use to users with limited programming experience. However, it did not work for me (under Ubuntu 14.04, and R 3.3.0, 64 bits). Even when running the Beta-Diversity.R script as a superuser I got the following error message.


> if (!require("ade4")) {install.packages("ade4")}
Loading required package: ade4
Installing package into ‘/usr/local/lib/R/site-library’
(as ‘lib’ is unspecified)
Error in contrib.url(repos, type) :
trying to use CRAN without setting a mirror


In addition to this, some of the required packages (such as “randomcoloR”) depended on system libraries which were not present in Ubuntu by default (libcurl, libv8). I was able to manually install all the unmet dependencies and run the full pipeline. However, since this manuscript emphasizes on Rhea being user friendly, this issue should be fixed, or at least generate an error message with further instructions that can be clearly interpreted by a newbie user.


- L111: “For the purpose of demonstrating and illustrating the different features of Rhea, the publicly available sequence data from the study by Müller (Müller et al. 2016), with ENA accession PRJEB13041, were analysed with the web platform IMNGS (www.imngs.org) and the output OTU-table and files (also available for download through the GitHub repository) were used for analysis. In this template study, the impact of housing conditions and diet on the mouse faecal microbiota and gut barrier was investigated.”

I have not been able to find any description on how the IMNGS platform works exactly, since the webpage (https://www.imngs.org/static/files/imngsPublication.pdf) only shows a “manuscript in preparation” message (as of Aug, 25, 2016). I see that the IMNGS manuscript is under review on another journal, but as it stands now no information is available on how the data used in this study was obtained. The authors should include a brief description of the IMNGS pipeline, perhaps as a supplementary material.

- L124: “Meticulous documentation of all scripts is provided online at the link given above. To minimize manual handling of data, intermediate files generated during processing are automatically transfer to folders where they are needed for downstream analysis, on the condition that the original folder structure of Rhea is kept unchanged.”

Typo: “automatically transferred” instead of “automatically transfer”.
The authors have indeed performed an outstanding job on comprehensively documenting their pipeline, and even the source code is actually pleasant to look at. However, Rhea forces the user to either modify the source code each time he wants to run it on a different file, or to change the name of his files so that they are compliant with Rhea's naming conventions. I believe that Rhea would benefit if the scripts were modified so that the different parameters can be passed from the command line instead.

As a minor detail regarding the documentation, I was unable to find the meaning on the “d=0.2” written at the top-right corners of the beta-diversity plots.

Also, the pdf documentation for the serial group comparisons script (specifically the section about the create_input_table.R script) appears to be incomplete.

- L139: “Since rarefying was shown to introduce unnecessary variations and be overall worse than simple normalization to proportions (obtained by dividing OTU-specific sequence counts per one sample by the sum of reads across all OTUs for the given sample) (McMurdie & Holmes 2014), we only calculate the latter in Rhea. The resulting proportions table can be turned back to non-integer normalized sequence counts via multiplication by a common denominator (we use the minimal sum of sequences across all samples).”

This reference to McMurdie & Holmes (2014) is in my opinion extremely misleading, as it gives the impression that normalization to proportions is in fact supported by the referenced paper. To the best of my understanding, this is far from being true. See the following excerpts from McMurdie & Holmes (2014):

“[...] As illustrated by this simple example, it is inappropriate to compare the proportions of OTU i, pi=kij/sj, without accounting for differences in the denominator value (the library size, sj) because they have unequal variances.”

“[...] In this sense alone, the random step in rarefying is unnecessary. Each count value could be transformed to a common-scale by rounding kij*smin/sj. Although this common-scale approach is an improvement over the rarefying method here defined, both methods suffer from the same problems related to lost data.”

“In our simulations, both rarefied counts and sample proportions resulted in an unacceptably high rate of false positive OTUs.”

McMurdie & Holmes (2014) do actually recommend the usage of R packages such as edgeR or DeSeq2 for the normalization of OTU counts; those could be easily included in the Rhea pipeline. The authors should either follow the recommendations of McMurdie & Holmes (2014), or explicitly discuss why they believe that normalization to proportions is enough for their purposes.

- L204: “Because the latter? is commonly regarded as the most robust
unconstrained ordination method in community ecology.”

Typo.

- Other comments:

1) When running the Rhea pipeline with the provided example data I had to manually change the value of the “group_name” or “independant_variable_name” variables from “Genome” to “Diet”, as the field “Genome” was not present in the example data. The default parameter values should be changed so that the example data can be processed with minimal user intervention, so that unexperienced users can quickly verify that the software is working for them.

2) I believe that the adoption of Rhea by non-expert users would be greatly enhanced if tutorials and wrapper scripts for running Rhea on the outputs of mothur, QIIME and UPARSE were included. Right now users are forced to use IMNGS or to write their own pre-processing scripts.

3) The authors state that they plan to continue updating Rhea with new features. I would like to encourage them to release the “paper-accepted” version of Rhea as version 1.0.0, and follow semantic versioning (http://semver.org/) in their subsequent releases.

4) Rhea is currently available under a GPL license. A BSD license, while still being an open-source license, might be preferable for academic software, as it facilitates its adoption by other developers (see for example http://ivory.idyll.org/blog/2015-on-licensing-in-bioinformatics.html).

---

## Round 0.2 · Minor Revisions

Dear authors

Thanks you for considering most comments raised by the reviewers. However, I invite you to solve some minor issues concerning installation and execution of some commands to help users using your pipeline.

·

Basic reporting

No comments

Experimental design

No comments

Validity of the findings

No comments

Additional comments

- Auto-install of dependencies in Ubuntu still fails ("error in contrib.url(repos, type) : trying to use CRAN without setting a mirror)
- The "starting from mothur" section on the tutorial is perhaps a bit too convoluted. A script that generates the Rhea-formatted input files from the outputs of mothur's make.shared and classify OTU would be helpful to users.

---

## Round 0.3 · accepted · Accept

Thanks for taking into account the minor comments raised by the reviewer